# Zero-1-to-G: Taming Pretrained 2D Diffusion Model for Direct 3D Generation

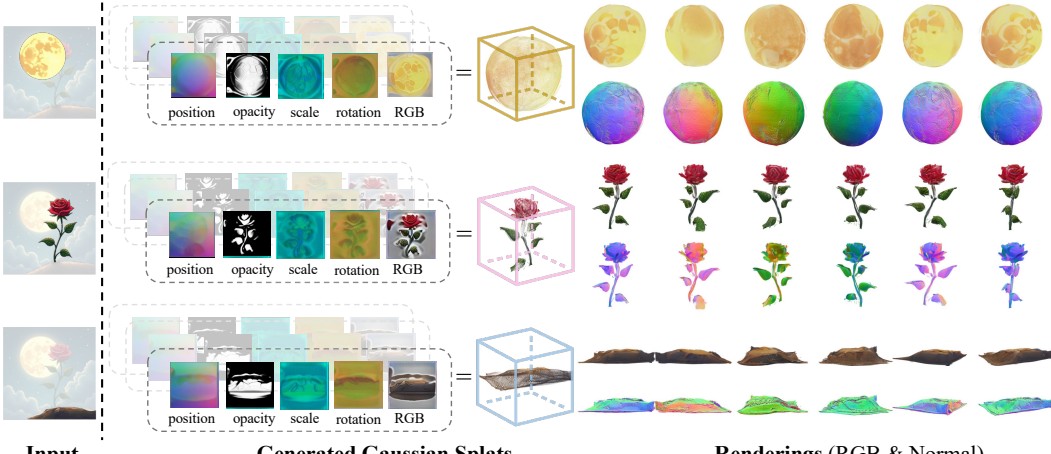

**Input**  **Generated Gaussian Splats**  **Renderings** (RGB & Normal)

Figure 1: **Zero-1-to-G** tackles direct Gaussian splat generation from single images. By using pretrained 2D diffusion models, we are able to generalize to in-the-wild objects.

## Abstract

Recent advances in 2D image generation have achieved remarkable quality, largely driven by the capacity of diffusion models and the availability of large-scale datasets. However, direct 3D generation is still constrained by the scarcity and lower fidelity of 3D datasets. In this paper, we introduce *Zero-1-to-G*, a novel approach that addresses this problem by enabling direct single-view **image-to-3D** generation on Gaussian splats using **pretrained** 2D diffusion models. Our key insight is that Gaussian splats, a 3D representation, can be decomposed into multi-view images encoding different attributes. This reframes the challenging task of direct 3D generation within a 2D diffusion framework, allowing us to leverage the rich priors of pretrained 2D diffusion models. To incorporate 3D awareness, we introduce cross-view and cross-attribute attention layers, which capture complex correlations and enforce 3D consistency across generated splats. This makes *Zero-1-to-G* the first direct 3D generative model to effectively utilize 2D pretrained diffusion priors, enabling efficient training and improved generalization to unseen objects. Extensive experiments on both synthetic and in-the-wild datasets demonstrate superior performance in 3D object generation, offering a new approach to high-quality 3D generation.

## 1 Introduction

Single image to 3D generation is a pivotal challenge in computer vision and graphics, supporting various downstream applications such as virtual reality and gaming technologies. A primary difficulty lies in managing the uncertainty of unseen regions, as these areas represent a conditional distribution based on the visible portions of a 3D object. Recent advancements in diffusion models (Ho et al., 2020; Rombach et al., 2022) have demonstrated significant efficacy in capturing complex data distributions within images and videos, prompting researchers to harness these models for single image to 3D generation. Early efforts distilled 3D neural fields from pretrained 2D diffusion models via score distillation (Poole et al., 2022; Wang et al., 2023). However, these approaches necessitate

per-scene optimization, which is time-consuming and susceptible to multi-faced Janus problems. Subsequent research achieved feed-forward generation by fine-tuning pretrained models to generate multi-view images of the same object (Liu et al., 2023b; Shi et al., 2023a; Long et al., 2023; Liu et al., 2023c) and enabling indirect 3D generation through sparse-view reconstruction models (Li et al., 2023; Tang et al., 2024; Xu et al., 2024), and Cat3D Gao et al. (2024) further extends sparse view generation to dense view generation for better reconstruction. Although these two-stage methods enhance quality and efficiency, they often yield poor geometric fidelity and blurry renderings due to inconsistencies in multi-view images. To circumvent these limitations, recent methodologies have trained diffusion models directly on 3D representations (Liu et al., 2023d; Chen et al., 2023a; Zhang et al., 2024a; He et al., 2024; Nichol et al., 2022), thereby eliminating the reliance on multi-view images. However, direct 3D generation techniques necessitate training from scratch, requiring substantial computational resources and large 3D datasets, which remain scarce—three orders of magnitude less prevalent than 2D data.

In this paper, we propose a novel approach for direct 3D generation that unites the strengths of both worlds: it leverages the expressive power of 2D diffusion networks while maintaining the 3D structural consistency required for accurate 3D generation. Our key contribution is bridging the gap between Gaussian splats and natural images typically used in 2D generation tasks. While the original Gaussian splats consist of 14-channel images encoding various attributes, we decompose each of them into multiple 3-channel attribute images while preserving its 3D information (Sec. 3.1) This decomposition also enables efficient latent diffusion training by projecting the splatter images into the latent space of a pretrained VAE, making our method directly generate 3D structures within a pretrained 2D diffusion framework. To ensure strong 3D consistency, we introduce cross-view attention layers in Stable Diffusion to enable information exchange between different viewpoints and cross-attribute attention mechanisms to maintain coherence across Gaussian attributes within a splatter image (Sec. 3.2). Additionally, we fine-tune the VAE decoder to address the domain gap between splatter images and natural images, as we observed that splatter image quality is highly sensitive to pixel-level variations (Sec. 3.3). By leveraging pretrained priors, our method not only generalizes better to unseen objects but also improves training efficiency compared to existing 3D generation methods.

It is important to note that, although we generate multiview splatter images, our method produces more structurally consistent and higher-quality results compared to traditional two-stage multiview 3D generation approaches. In two-stage methods (Xu et al., 2024; Tang et al., 2024), strong pixel-level consistency is required in the first stage to ensure accurate reconstruction in the second stage, which is often difficult to achieve. Moreover, the first stage operates independently of the second, lacking coordination between the two, thereby exacerbating inconsistencies. In contrast, our approach directly generates the final 3D representation in a single stage, eliminating the need for pixel-level correspondence between multiview splatter images. Since splatter images can contain redundant information, and their spatial positions do not necessarily map to the final 3D positions, this single-stage process offers greater flexibility and robustness, ensuring a more consistent final 3D structure.

Overall, our contributions can be summarized as below:

- We present *Zero-1-to-G*, a novel direct 3D generative model for Gaussian splats that achieves excellent 3D consistency and superior rendering quality.

- We observe that Gaussian splats, as a 3D representation, can be decomposed into multi-view images representing different attributes, making them inherently compatible with backbones designed for 2D image generation.

- Through splatter image decomposition, we unleash the power of pretrained 2D diffusion models for direct 3D generation, enabling us to complete training more efficiently and have better generalization towards in-the-wild data.

## 2 RELATED WORKS

**Optimization-based 3D Generation** Dreamfusion (Poole et al., 2022) and subsequent works (Lin et al., 2023; Chen et al., 2023b; Wang et al., 2023) utilize a pretrained text-to-image diffusion model to optimize a 3D representation through score distillation. DreamGaussian (Tang et al., 2023) sig-

nificantly reduces training time by optimizing Gaussian splats. However, score distillation-based methods still require minutes of optimization per scene, as they must compare renderings with diffusion outputs from various viewpoints, which limits their generation speed. Additionally, these methods lack a clear understanding of geometry and viewpoint, resulting in multi-face problems.

**Direct 3D Generation** Significant efforts have been made to directly train diffusion models on various 3D representations, including point clouds (Luo & Hu, 2021; Zhou et al., 2021; Nichol et al., 2022; Jun & Nichol, 2023), meshes (Liu et al., 2023d), and neural fields (Chen et al., 2023a; Shue et al., 2023; Müller et al., 2023). However, these methods are typically constrained to category-level datasets and often struggle to generate high-quality assets. More recent approaches have begun encoding 3D assets into more compact latent representations (Zhang et al., 2023a; Lan et al., 2024; Zhao et al., 2023; Zhang et al., 2024b; Hong et al., 2024; Dong et al., 2024; **?**), enabling diffusion models to be trained more efficiently and enhancing generalization capabilities. Despite these advancements, direct 3D generative models are still primarily trained on synthetic 3D datasets like Objaverse (Deitke et al., 2024), which may hinder their ability to effectively handle more in-the-wild inputs.

More closely related to our work are GVGen (He et al., 2024) and GaussianCube (Zhang et al., 2024a), which investigate training diffusion models on Gaussian splats. Both approaches acknowledge the challenges of directly learning diffusion models from Gaussian splats and propose organizing the Gaussian points into a more structured volume. Different from their methods, we adopt a multi-view splatter image representation, enabling us to train diffusion models on Gaussian splats directly with high efficiency.

**Finetuning Pretrained Diffusion models** To enhance the 3D awareness of 2D pretrained diffusion models, MVDream (Shi et al., 2023b)integrates cross-view attention layers and fine-tunes them to produce multi-view renderings. As an application, LGM (Tang et al., 2024) and InstantMesh (Xu et al., 2024) utilize a pre-trained single-view to multi-view 2D diffusion model, transforming the single view generation problem to a multi-view reconstruction task. These methods adopt various compact 3D representations to adapt to the sparse view 2D input. This has greatly pushed the 3D generation to more complex data. However, this approach is constrained by the performance of the multi-view diffusion model, which often lacks strict multi-view consistency. This inconsistency can result in poor geometry and blurry textures in the final 3D reconstructions.

Some other works further leverage the power of pretrained diffusion models to generate data beyond the domain of natural images. For instance, JointNet (Zhang et al., 2023b) uses two diffusion models for joint RGB and depth prediction. Marigold (Ke et al., 2023) fine-tunes pretrained diffusion models for monocular depth estimation from single image input. Wonder3D (Long et al., 2023) generates both multi-view RGB and normal maps from a unified diffusion model equipped with a domain switcher. Similarly, GeoWizard (Fu et al., 2024) predicts depth and normals from a single image using cross-domain geometric self-attention to maintain geometric consistency. Following this line of work to adapting diffusion prior to images of other domains, our method aims to generate Gaussian splats represented as splatter images using pretrained 2D diffusion models, thereby enhancing the ability of direct 3D generation to tackle in-the-wild images.

**3D Generation with Reconstruction-based methods** Researchers have opted to train reconstruction-based models for highly efficient 3D generation Hong et al. (2023); Tochilkin et al. (2024); Woo et al. (2024); Zou et al. (2023); Xu et al. (2023). LRM (Hong et al., 2023) and TripoSR (Tochilkin et al., 2024) introduced a transformer-based model that directly output a triplane from a single image. The model is trained on million-scale data by comparing the renderings of the triplane with ground truth using regression-based loss. TriplaneGaussian (Zou et al., 2023) further used a hybrid triplane-Gaussian representation to greatly accelerate the rendering of the generated 3D assets. However, the main drawback of regression-based methods is their failure to account for the uncertainty in single-view to 3D generation. Although GECO (Wang et al., 2024) attempts to address this issue by distilling knowledge from multi-view diffusion models into a feedforward model, its results remain limited by the quality of the generated multi-view images.

**Direct 3D Generation with 2D Diffusion** Concurrent works Yan et al. (2024) and Elizarov et al. (2024) also propose methods of generating 3D with 2D diffusion models. They use UV atlas to encode the geometry and texture of 3D mesh, while we use Splatter Image Szymanowicz et al. (2023) as 3D representation. Since Omage Yan et al. (2024) uses a 12-channel UV atlas, its model

cannot utilize the pretraiend 2D diffusion prior and has to be trained from scratch, limiting its ability within the distribution of the training dataset. On the other hand, GIMDiffusion Elizarov et al. (2024) decomposes the UV atlas into separate geometry maps and albedo textures to match the 3-channel output of pretraiend 2D diffusion model. Still, they tackle the task of text-to-3D generation while we focus more on single-view image-to-3D reconstruction.

**Direct 3D Generation with 2D Diffusion** Concurrent works, Omage Yan et al. (2024) and GIMDiffusion Elizarov et al. (2024), also explore the utilization of 2D diffusion models for 3D generation but differ from ours in the representation and model architecture. Omage Yan et al. (2024) employs a 12-channel UV atlas to encode the whole 3D meshes, thus precluding the use of pretrained 2D diffusion models. As a result, Omage requires training from scratch on category-level datasets, confining its generative capability to the distribution of its training data. In contrast, GIMDiffusion Elizarov et al. (2024) adopts a similar idea of using pretrained 2D diffusion models through decomposition, but the utilization of pretrained 2D prior is only limited to the generation of albedo texture: it consists of a frozen pretrained model for albedo texture generation, and another architectural clone trained from scratch for geometry generation.

In our work, we leverage Splatter Image Szymanowicz et al. (2023) as the 3D representation, with our insight that each attribute of the Splatter Image is modeled within the distribution of pretrained 2D diffusion, therefore using the diffusion prior to generate the entire 3D representation.

## 3 METHODS

Our method Zero-1-to-G is a single stage direct 3D generation: given single view input $\mathbf{I}$, Zero-1-to-G generates the corresponding 3D representation $\mathbf{z}$, where $z = \{z_i | i = 1, ..., N\}$ multiple splatter images under $\mathbf{N}$ camera views.

To harness the power of large-scale pretrained 2D diffusion models for direct 3D generation, we represent each 3D object as a set of multi-view splatter images (Szymanowicz et al., 2023). In Sec. 3.1, we detail our decomposition process, converting each multi-view splatter image into five 2D attribute images corresponding to RGB color, scale, rotation, opacity, and position. This decomposition allows us to effectively leverage the priors of 2D pretrained diffusion models to learn the underlying 3D object distribution (Sec. 3.2). Furthermore, we fine-tune the VAE decoder to enhance the rendering quality of the decoded splatter images (Sec. 3.3).

### 3.1 REPRESENTING 3D OBJECTS AS 2D IMAGES

A splatter image is a rearrangement of Gaussian splat into regular grids of $H \times W$, with 14 channels stacking 5 Gaussian splat attributes, including 3-channel RGB, 4-channel rotation, 3-channel scale, 3-channel position, and 1-channel opacity. By applying the data transformation in A.1, each attribute can be represented as a 2D image that explicitly models the object's appearance or geometry. Our key observation is that, each attribute image is well modeled within the distribution of pretrained 2D diffusion model, which means the pretrained 2D diffusion prior might be helpful in the generation of all these attribute images. Motivated by previous works that leverage pretrained diffusion priors for domain-specific images that contain rich 3D information such as normal or depth (Ke et al., 2023; Long et al., 2023; Fu et al., 2024), we take a step further and generate five attribute images of Gaussian splats. Since StableDiffusion is trained on image data in the range $[-1, 1]$, we normalize the values of different attributes to ensure compatibility. Detailed operation is included in the appendix A.1. After applying the data transformations, we convert the 14-channel splatter into a set of 5 multi-view, 3-channel attribute images that are normalized to the RGB space. Example visualization of multi-view splatters are shown in Figure 2.

To obtain our splatter image ground truth, we fine-tune LGM (Tang et al., 2024) to generate Gaussian splats using six views of 3D object renderings as input. A straightforward way would be to directly fit Gaussian splats from multi-view renderings of each object. However, we found that this direct fitting results in attribute images with excessive high-frequency signals compared to LGM output (Figure 3), which makes it challenging for the pretrained VAE to decode effectively.

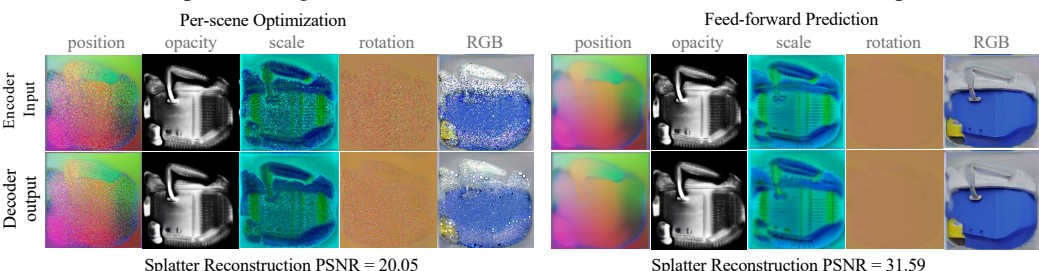

Figure 2: The pipeline of Zero-1-to-G. During training, we fine-tune both the VAE decoder and the denoising UNet of Stable Diffusion. Decoder fine-tuning is required for high-quality splatter image rendering because the renderings of splatter images are sensitive to pixel value changes; the attention mechanism of denoising UNet is extended to model the multi-view and multi-attribute correlation. At inference time, given a single view input of the target object, each component in the splatter image is generated by conditioning the camera view and attribute switcher. The generated set of splatter image components can be directly composed into Gaussian splats. Note that here we only show 3 views of splatter images for better visualization, but we use 6 views in our experiment.

Figure 3: VAE encoding and decoding comparison with per-scene optimized splatters and feed-forward predicted splatters.

## 3.2 DIRECT 3D GENERATION VIA 2D PRETRAINED DIFFUSION

With the decomposition discussed in Sec 3.1, we are ready to directly learn a 3D generative model on Gaussian splats represented by a set of multi-view attribute images. The distribution of our objects represented as splatter images, denoted as $p(\mathbf{z})$, is modeled as a joint distribution of 6 fixed camera views and 5 splatter attributes. Given a set of fixed camera viewpoints $\{\boldsymbol{\pi}_1, \boldsymbol{\pi}_2, \cdots, \boldsymbol{\pi}_K\}$ and condition input image $y$:

$$p(\mathbf{z}) = p(\mathbf{z}^{(1:K,1:N)}|y) = p_{\text{pos, op, sc, rot, rgb}}\left(\{\mathbf{z}_{\text{pos}}^{1:K}, \mathbf{z}_{\text{op}}^{1:K}, \mathbf{z}_{\text{sc}}^{1:K}, \mathbf{z}_{\text{rot}}^{1:K}, \mathbf{z}_{\text{rgb}}^{1:K}\} \mid y\right) \quad (1)$$

where {pos, op, sc, rot, rgb} are the 5 attributes of the splatter image.

Namely, we can naturally utilize the pretrained 2D diffusion models to generate these attribute images since they are already formatted as 3-channel images. However, original Stable Diffusion can only generate single-view images, while our goal is to learn the joint distribution across multiple views and attributes. To ensure that the generated 5 multi-view attribute images are coherent and represent the same object, we insert additional attention layers into the pretrained diffusion UNet to jointly model both the cross-view and cross-attribute distribution of our decomposed splatter images.

**Modeling Multi-view Distribution** Prior works have approached multi-view diffusion either by reshaping the batch dimension into a token dimension and applying self-attention (Shi et al., 2023b; Liu et al., 2023c; Long et al., 2023; Liu et al., 2024), or by spatially concatenating multi-view images to form a larger image, which directly maps the latent distribution to a multi-view distribution (Shi et al., 2023a). We choose the former approach for its flexibility in reshaping data for both cross-view and cross-attribute attention mechanisms. This design allows for efficient information exchange among different views, where tokens corresponding to the same attribute from different views are concatenated for self-attention. This facilitates our model's ability to learn a consistent multi-view distribution for each Gaussian attribute.

**Modeling Multi-Attribute Distribution** Building on the work of (Long et al., 2023), we utilize an attribute switcher to specify which attribute the network should generate. To maintain consistency across generated images that represent different attributes of the same object, we employ an attention mechanism to capture the interactions between images taken from the same viewpoint but corresponding to different attributes.

Specifically, we introduce additional self-attention modules to model the cross-attribute correlations, where tokens representing all attributes from the same viewpoint are combined and processed using standard scaled dot-product attention.

**Training Loss** During training, we organize each view and attribute of a splatter image within the batch dimension and apply independently sampled Gaussian noise. In each attention block, we alternately apply multi-view attention and multi-attribute attention to enhance the model's ability to learn complex correlations.

The forward process of our diffusion model is directly extended from the original DDPM (Ho et al., 2020), which is

$$q(\mathbf{z}_{1:T}^{(1:K,1:N)}|\mathbf{z}_0^{(1:K,1:N)}) = \prod_{t=1}^{T} q(\mathbf{z}_t^{(1:K,1:N)}|\mathbf{z}_{t-1}^{(1:K,1:N)}) = \prod_{t=1}^{T}\prod_{k=1}^{K}\prod_{n=1}^{N} q(\mathbf{z}_t^{(k,n)}|\mathbf{z}_{t-1}^{(k,n)}), \quad (2)$$

And the reverse process will be

$$p_\theta(\mathbf{z}_{0:T}^{(1:K,1:N)}) = p(\mathbf{z}_T^{(1:K,1:N)}) \prod_{t=1}^{T} p_\theta(\mathbf{z}_{t-1}^{(1:K,1:N)}|\mathbf{z}_t^{(1:K,1:N)}) \qquad (3)$$

$$= p(\mathbf{z}_T^{(1:K,1:N)}) \prod_{t=1}^{T}\prod_{k=1}^{K}\prod_{n=1}^{N} p_\theta(\mathbf{z}_{t-1}^{(k,n)}|\mathbf{z}_t^{(1:K,1:N)}), \qquad (4)$$

where $p_\theta(\mathbf{z}_{t-1}^{(k,n)}|\mathbf{z}_t^{(1:K,1:N)}) = \mathcal{N}(\mathbf{z}_{t-1}^{(k,n)}; \mu_\theta^{(k,n)}(\mathbf{z}_t^{(1:K,1:N)}, t), \sigma_t^2\mathbf{I})..$ The definition of the Gaussian mean for the reverse process is defined as:

$$\mu_\theta^{(k,n)}(\mathbf{z}_t^{(1:K,1:N)}, t) = \frac{1}{\sqrt{\alpha_t}}\left(\mathbf{z}_t^{(k,n)} - \frac{\beta_t}{\sqrt{1-\bar{\alpha}_t}}\epsilon_\theta^{(k,n)}(\mathbf{z}_t^{(1:K,1:N)}, t)\right), \qquad (5)$$

The corresponding loss function for multi-view and multi-domain modeling is as follows:

$$\ell = \mathbb{E}_{t,\mathbf{x}_0^{(1:K,1:N)},k,n,\epsilon^{(1:K,1:N)}}\left[\|\epsilon^{(k,n)} - \epsilon_\theta^{(k,n)}(\mathbf{z}_t^{(1:K,1:N)}, t)\|_2^2\right], \qquad (6)$$

where $\epsilon^{(k,n)}$ is the Gaussian noise added to attribute $n$ for the $k$-th view, and $\epsilon_\theta^{(k,n)}$ is the model's noise prediction for attribute $n$ in the $k$-th view.

## 3.3 VAE DECODER FINE-TUNING

The pretrained VAE of Stable Diffusion is initially trained on natural images. While directly utilizing this VAE can produce visually appealing splatter images, the 2D renderings often exhibit noticeable artifacts. This issue arises from two main factors: (1) each pixel in the splatter image corresponds to a Gaussian splat, meaning that even minor changes in pixel values can significantly impact the final rendering, and (2) splatter images contain high-frequency details that are challenging for the VAE to recover accurately.

To leverage the pretrained knowledge while enhancing rendering quality, we freeze the VAE encoder and only fine-tune the decoder. This strategy preserves the latent space of the pretrained model. In addition to using the standard reconstruction loss based on the direct decoder output, we also evaluate the rendering quality of the decoded splatter by comparing its renderings with ground truth images. This rendering-based loss is essential, as a low reconstruction error in the splatter image does not guarantee high-quality renderings. The rendering loss is composed of MSE loss and LPIPS loss, defined as follows:

$$\mathcal{L}_{\text{rgb}} = \mathcal{L}_{\text{MSE}} + \mathcal{L}_{\text{LPIPS}} \tag{7}$$

The overall objective function of decoder finetuning is defined as:

$$\mathcal{L}_{\text{decoder}} = \mathcal{L}_{\text{splatter}} + \mathcal{L}_{\text{normal}} + \mathcal{L}_{\text{rgb}} + \mathcal{L}_{\text{mask}} \tag{8}$$

where $\mathcal{L}_{\text{splatter}}$ denotes the reconstruction loss of splatter image itself, while $\mathcal{L}_{\text{normal}}$, $\mathcal{L}_{\text{rgb}}$ and $\mathcal{L}_{\text{mask}}$ are all about the renderings of the reconstructed splatter, denoting cosine similarity loss of rendered normals, the sum of all losses of rendered images, and the binary cross-entropy loss of the rendered masks.

## 4 EXPERIMENTS

### 4.1 IMPLEMENTATION DETAILS

**Dataset** We train on the G-buffer Objaverse (Qiu et al., 2024) dataset, which consists of approximately 262,000 objects sourced from Objaverse (Deitke et al., 2024). Each object in the dataset is rendered from 38 viewpoints, with additional normal and depth renderings provided. For generating the ground truth splatter images, we use the first viewpoint as the input condition, along with five additional views at the same elevation and azimuth angles of 30°, 90°, 180°, 270°, and 330° to comprehensively cover the full 360 degrees. We only use the RGB and normal renderings for the supervision of decoder finetuning and Gaussian Splats prediction model.

**Model Training and Inference** We initialize our model from Stable Diffusion Image Variations. Following Wonder3D (Long et al., 2023), our training includes two stages. In the first stage, we only train multi-view attention, and in the second stage, we add one more cross-domain attention layer for training, and together fine-tune the multi-view attention layer learned in the first stage. For the first stage, we use a batch size of 64 on 4 NVIDIA L40 GPUs for 13k iterations, which takes about 1 day. For the second stage, we use a batch size of 64 on 8 NVIDIA L40 GPUs for 30k iterations, which takes about 2 days. For decoder fine-tuning, we use a total batch size of 64 on 8 NVIDIA L40 GPUs for 20k iterations. The second stage of training takes about 2 days. During inference, we use cfg = 3.5 and our method can generate Gaussian splats per object in 8.7 seconds on a single NVIDIA L40 GPU.

### 4.2 EVALUATION PROTOCOL

**Dataset and Metrics** Following prior works (Liu et al., 2023c;a; Wang et al., 2024), we conduct our quantitative comparisons using the Google Scanned Object (GSO) dataset (Downs et al., 2022). We utilize a randomly selected subset of 30 objects from the GSO, encompassing a range of daily items and animals, as in SyncDreamer (Liu et al., 2023c). For each object, we render a condition image at a resolution of $512 \times 512$ and an elevation of $10°$. Additionally, we generate evenly sampled evaluation images at $30°$ azimuthal intervals around the object, maintaining the same elevation. We report commonly used metrics for novel view synthesis, including PSNR, SSIM (Wang et al., 2004), and LPIPS (Zhang et al., 2018). Beyond the GSO dataset, we also evaluate our methods on in-the-wild images.

**Baselines** We compare our methods against several recent approaches across different categories. For reconstruction-based methods, we include TriplaneGaussian (Zou et al., 2023) and TripoSR (Tochilkin et al., 2024). In the realm of direct 3D generation, we compare with LN3Diff (Lan et al., 2024). Finally, for two-stage methods transitioning from single-image to multi-view to 3D, we include InstantMesh (Xu et al., 2024) and LGM (Tang et al., 2024).

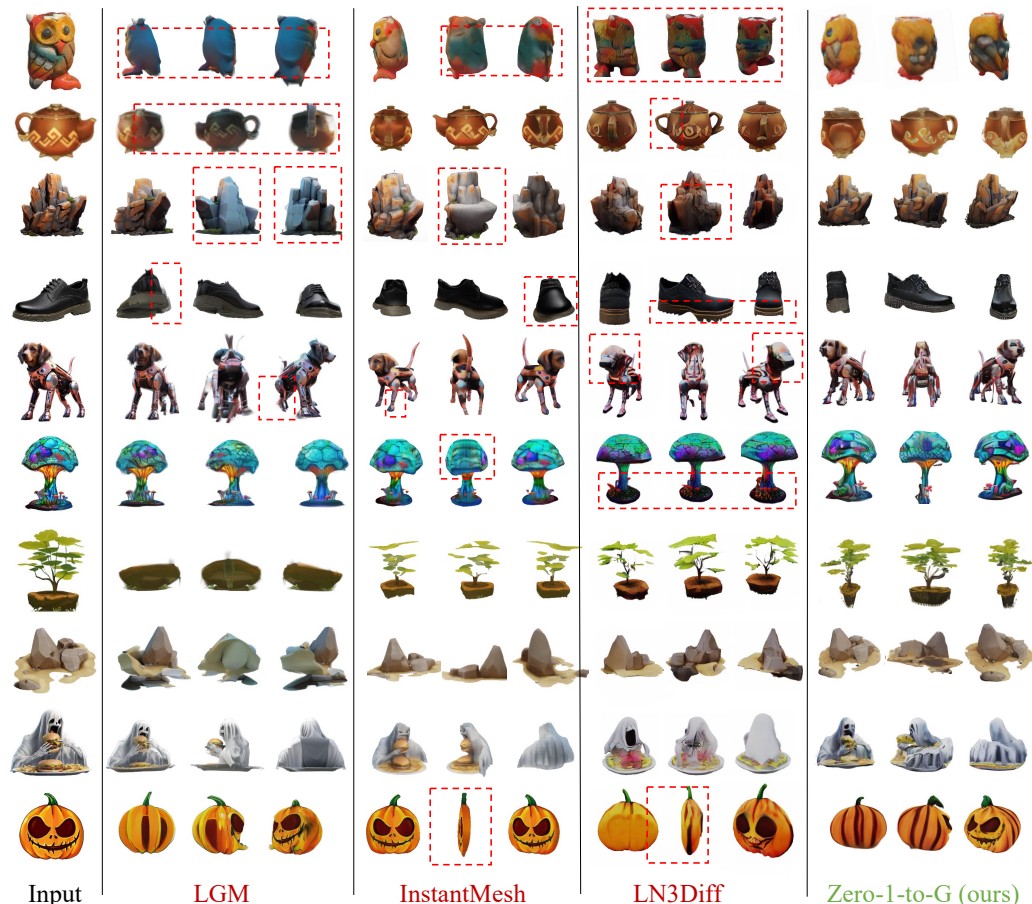

| Input | LGM | InstantMesh | LN3Diff | Zero-1-to-G (ours) |

Figure 4: Qualitative comparison with, LGM, InstantMesh, LN3Diff on in-the-wild data.

Table 1: Quantitative comparison between our methods and other baselines on the GSO dataset.

| Methods | PSNR ↑ | SSIM ↑ | LPIPS ↓ | CD ↓ |
|---|---|---|---|---|
| TriplaneGaussian (Tang et al., 2023) | 17.80 | 0.811 | 0.216 | 0.0440 |
| TripoSR (Tochilkin et al., 2024) | 17.32 | 0.804 | 0.217 | 0.0423 |
| LGM (Tang et al., 2024) | 17.01 | 0.793 | 0.199 | 0.0621 |
| InstantMesh (Xu et al., 2024) | 18.15 | 0.810 | 0.179 | 0.0419 |
| LN3Diff (Lan et al., 2024) | 16.30 | 0.786 | 0.241 | 0.0637 |
| Ours-fast(10 steps) | 19.03 | 0.812 | 0.182 | 0.0396 |
| Ours (50 steps) | **19.40** | **0.818** | **0.178** | **0.0390** |

## 4.3 RESULTS

**Qualitative Comparison** Figure 4 showcases the renderings produced by Zero-1-to-G alongside various baselines for in-the-wild image inputs. We find that two-stage methods relying on multi-view images are often limited by the quality of the generated views. For example, LGM frequently exhibits blue and overly smooth textures in the background (first and third examples in Figure 4), a result of artifacts in the multi-view images generated by ImageDream (Wang & Shi, 2023). Furthermore, their reconstruction model struggles to maintain view-consistent 3D Gaussians, leading to "floaters" around the object (fourth and fifth examples in Figure 4). While InstantMesh provides more consistent renderings, its outputs tend to be overly smooth, as it fuses inconsistent multi-view images into a triplane representation. And its texture also suffers from grid-like artifacts brought by its mesh generation process. Both LGM and InstantMesh are constrained by the limitations of upstream 2D multi-view image generation models, resulting in unreasonable geometry and view-inconsistency features (last example in Figure 4). In contrast, our method incorporates inherent 3D representations, yielding accurate geometry and view-consistent renderings.

Compared to previous direct 3D generation methods, LN3Diff (Lan et al., 2024) also faces challenges with complex textures and fails to produce high-quality reconstructions from single-view inputs. Their textures often lack fine-grained details, and the geometry tends to be oversmoothed. This issue arises because their model is trained from scratch, making it difficult to achieve high-quality 3D generation given the available computational resources and datasets. Consequently, this limits their generalization ability on unseen objects.

In contrast, by leveraging pretrained 2D diffusion priors, our approach delivers high-fidelity 3D renderings with accurate geometry and texture, effectively handling in-the-wild input images.

**Quantitative Comparison** The results of the quantitative comparison on the GSO dataset are presented in Table 1. Zero-1-to-G consistently outperforms all baselines across all metrics. Reconstruction-based methods, such as TriplaneGaussian and TripoSR, struggle to deliver sharp predictions for unseen regions due to their deterministic nature. Two-stage methods, like InstantMesh, demonstrate relatively good performance; however, they still fall short compared to our approach due to the limitations imposed by sparse multi-view images. In contrast, direct 3D methods such as LN3Diff fail to yield satisfactory results as they lack pretrained priors.

**Training Efficiency** By leveraging pretrained diffusion priors, our method reduces training time and resource requirements. We complete training in just 3 days using only 8 NVIDIA L40 GPUs, which is more efficient compared to previous two-stage and direct 3D generation methods, as detailed in Table 2. This efficiency highlights the advantage of integrating 2D priors for direct 3D generation, reducing the need for extensive computational resources.

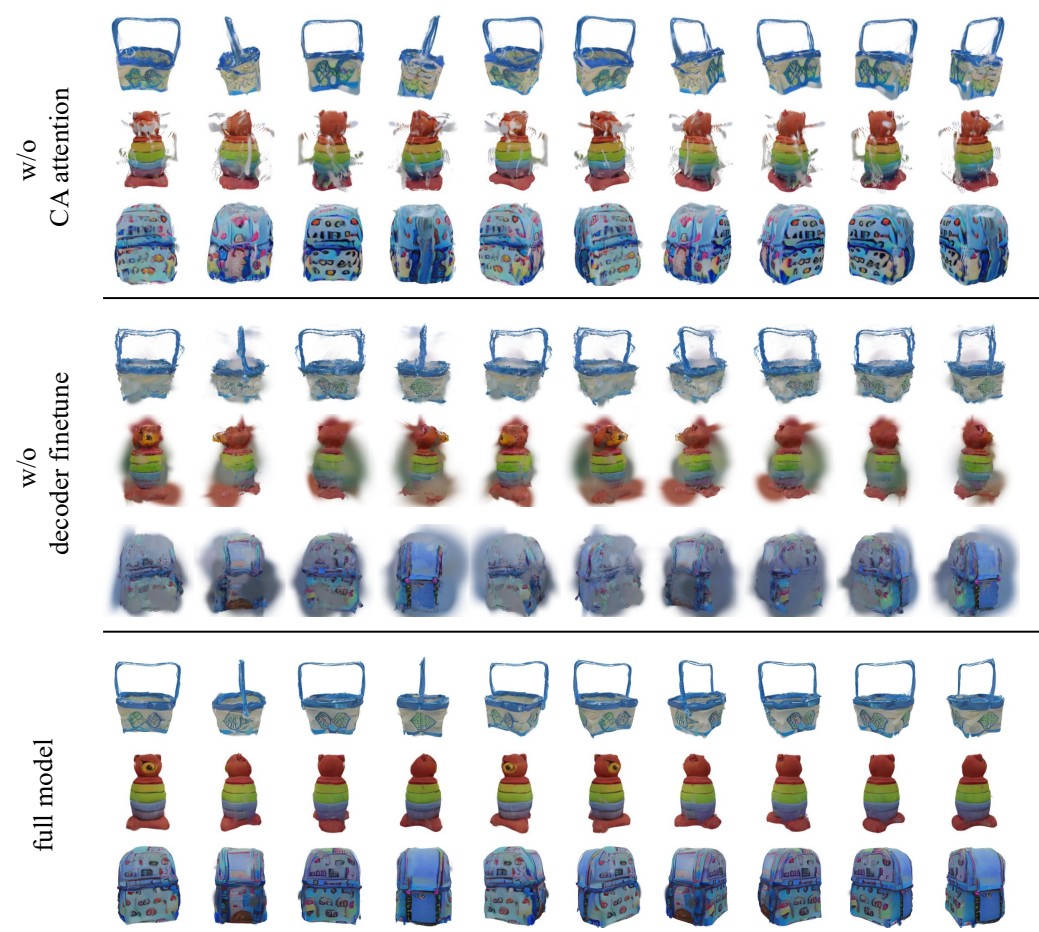

Figure 5: Ablation study on GSO dataset.

## 4.4 ABLATION STUDY

**VAE Decoder Finetuning** Without fine-tuning VAE decoder, although the decoded splatter image visually looks fine, the renderings exhibit noticeable artifacts. Since each pixel represents a Gaussian splat and the decoder cannot capture high-frequency areas, well-reconstructed splatter images don't necessarily ensure good renderings.

**Cross-attribute Attention** We can see from Figure 5, if we don't use cross-attribute attention, the renderings of the Gaussian splats have many floaters and the textures become blurry, this is because that different attributes of the same Gaussian splat are not well aligned.

**No Diffusion Prior** The use of diffusion prior is essential in our model. To verify this, we conducted training with the same StableDiffusion UNet architecture but using random initialization, and we also added the same cross-view attention layers to the UNet as in our method. We can see that without using the prior, the training of the model cannot converge to meaningful results using the same data and training iterations.

Table 2: Comparison of training efficiency with other baseline methods. For LGM and InstantMesh, we only count the reconstruction part. More time and resources are needed to train their multi-view generation model.

| Methods | Training Time ↓ | GPUs ↓ |
|---|---|---|
| LGM (reconstruction module) | 4 days | 32 * A100 (80G) |
| InstantMesh (reconstruction module) | 12 days | 16 * H800 (80G) |
| LN3Diff | 7 days | 8 * A100 (80G) |
| Ours | 3 days | 8 * L40 (48G) |

Table 3: Abalation study on module design, inference with GSO dataset.

| Model Components | PSNR ↑ | SSIM ↑ | LPIPS ↓ |
|---|---|---|---|
| w/o diffusion prior | 9.39 | 0.592 | 0.722 |
| w/o decoder fine-tuning | 17.13 | 0.775 | 0.272 |
| w/o cross-attribute attention | 17.26 | 0.767 | 0.237 |
| Full Model | **19.40** | **0.818** | **0.178** |

## 5 CONCLUSION

In this work, we introduce a novel framework that leverages 2D diffusion priors for direct 3D generation by decomposing Gaussian splats into multi-view attribute images. This decomposition preserves the full 3D structure while efficiently mapping it to 2D images, enabling fine-tuning of pre-trained Stable Diffusion models with cross-view and cross-attribute attention layers. Our approach significantly reduces computational costs compared to other direct 3D generation methods. By bypassing the stringent requirement for multi-view image consistency in two-stage approaches, we generate more accurate 3D geometry and produce higher-quality renderings through a single-stage diffusion process. Furthermore, our method exhibits stronger generalization capabilities than existing direct 3D generation techniques due to the use of diffusion priors, offering a more efficient and scalable solution for 3D content creation.

In future work, we plan to extend our approach to 4D generation by fine-tuning video models with dynamic Gaussian splats mapped to multi-view attribute images. While our method produces high-quality 3D results, it currently faces limitations in speed due to the multi-step denoising process, which results in slower inference speed. To address this, future work will explore consistency models as a potential solution to improve inference speed while maintaining the quality of 3D generation.

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

## A APPENDIX

### A.1 IMPLEMENTATION DETAILS

**Splatter Image Transformation**
Each attribute, except opacity, possesses three degree-of-freedoms, which align gracefully with the 3 channels of the RGB space. The following illustrates the detailed operations to convert each attribute into an RGB image

- RGB: the RGB attribute naturally lies in the RGB space, no conversion is needed.
- Position: We normalize the 3D object in the bounding box $[-1, 1]$ and use the 3D coordinates of each Gaussian as position attribute.
- Scale: The raw scale value spans from $1e-15$ to $1e-2$, so directly converting the 3D scale to RGB space using the min-max value for the whole dataset will result in most regions being zeros due to the significant difference in power. The value distribution also does not match the normal distribution, making it difficult for diffusion models to learn effectively. We thus convert the raw scale values to log-space and clamp the minimum values to $-10$, as we found Gaussian splats with scales smaller $1e-10$ will have negligible effects on the final rendering.
- Rotation: We first convert the 4-dimensional quaternion to 3-dimensional axis angle, then normalize it to $[-1, 1]$.
- Opacity: We directly duplicate the single channel to 3 channels, and average the predicted 3-channel image to get the final opacity prediction.

**UNet Fine-tuning** When fine-tuning the StableDiffusion UNet, for both stages, we use a constant learning rate of $1e-4$ with a warmup of the first 100 steps. We use the Adam optimizer for both stages and the betas are set to $(0.9, 0.999)$. For classifier-free guidance, we drop the condition image with a probability of $0.1$.

**LGM Fine-tuning** To obtain the splatter image ground truth for our training, as mentioned in Sec. 3.1, we fine-tune LGM (Tang et al., 2024) to take as input 6 multi-view renderings of the G-Objaverse dataset and output splatter images of 2D Gaussian splatting (Huang et al., 2024). The training objective is to compare the splatter renderings with ground truth images using MSE and LPIPS loss. We also use cosine similarity loss between ground truth normals and rendered normals. We fine-tune LGM for 30k iterations with a batch size of 32 on 8 NVIDIA L40 GPUS, which takes about 1 day.

### A.2 LIMITATIONS

Despite achieving superior reconstruction metrics and strong generalization to in-the-wild data, our method has some limitations. First, our inference speed is not as fast as regression models, as each splatter must be generated through diffusion. A potential improvement would be to integrate a consistency model Song et al. (2023) to reduce denoising steps and enhance inference speed. Second, we do not currently disentangle material and lighting conditions, leading to highlights and reflections being baked into the Gaussian splat texture. Future work could address this by incorporating inverse rendering to better predict non-Lambertian surfaces.

## A.3 MORE RESULTS

**Chamfer Distance** We have further evaluated Chamfer Distance on the GSO datasets to show the superiority of our generated geometry. The improvement in Chamfer Distance further validates our claim that our approach generates superior 3D representations, demonstrating the effectiveness of our method.

Table 4: Comparison of Chamfer Distance across methods.

| Methods | Chamfer Distance ↓ |
|---|---|
| LGM Tang et al. (2024) | 0.0621 |
| InstantMesh Xu et al. (2024) | 0.0419 |
| LN3Diff Lan et al. (2024) | 0.0637 |
| Ours | **0.0390** |

**More visualizations on ablation** Figure 6 shows the splatter image visualization of ablation study. We can see that without cross-attribute attention, there are obvious misalignments of different domains of the splatter images. Without decoder fine-tuning, although the splatter image is visually good, the rendering is not satisfying because Gaussian splats are sensitive to the value changes in the pixels. Fine-tuning the decoder can greatly improve the rendering quality.

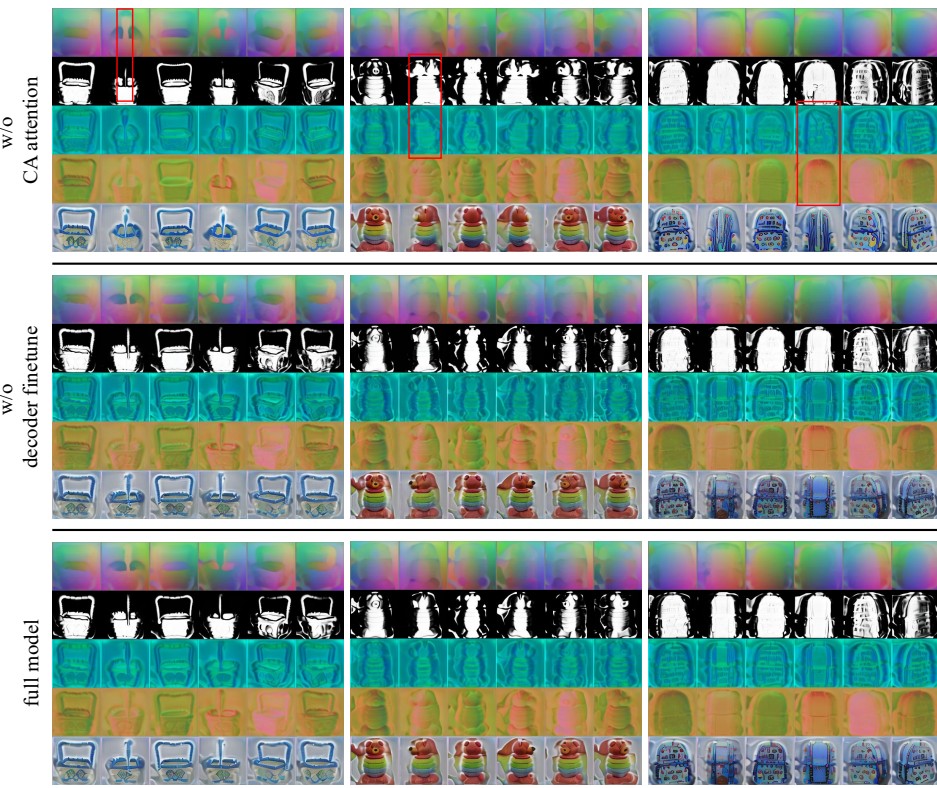

Figure 6: Splatter visualization of ablation study.

**More examples** More RGB and normal renderings can be found in Figure 7.

**Diversity** Since we model the unseen viewpoints with diffusion models, our results can generate diverse results given the same input Figure 8.

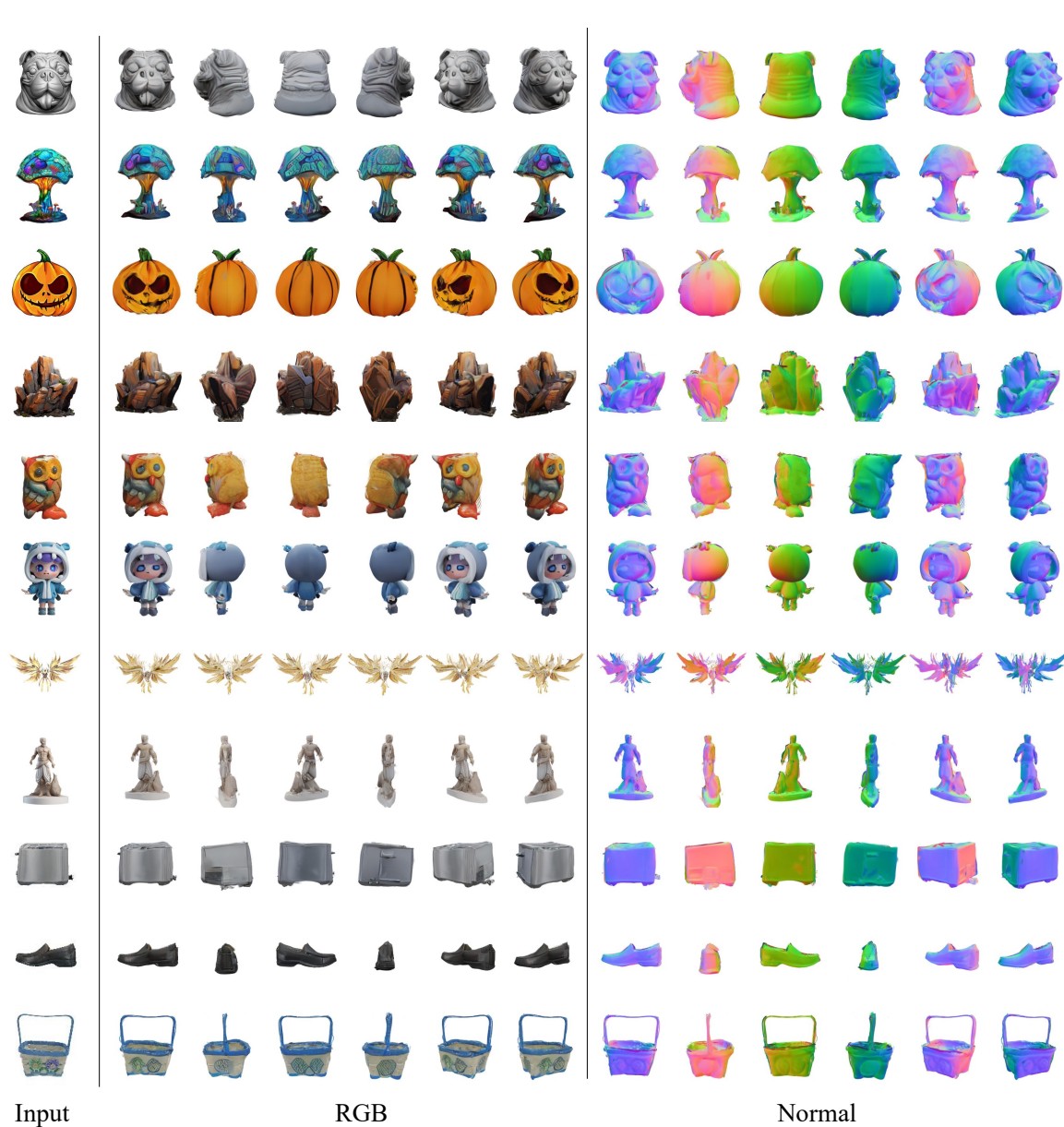

Input           RGB           Normal

Figure 7: RGB and normal renderings of more examples on in-the-wild and GSO datasets.

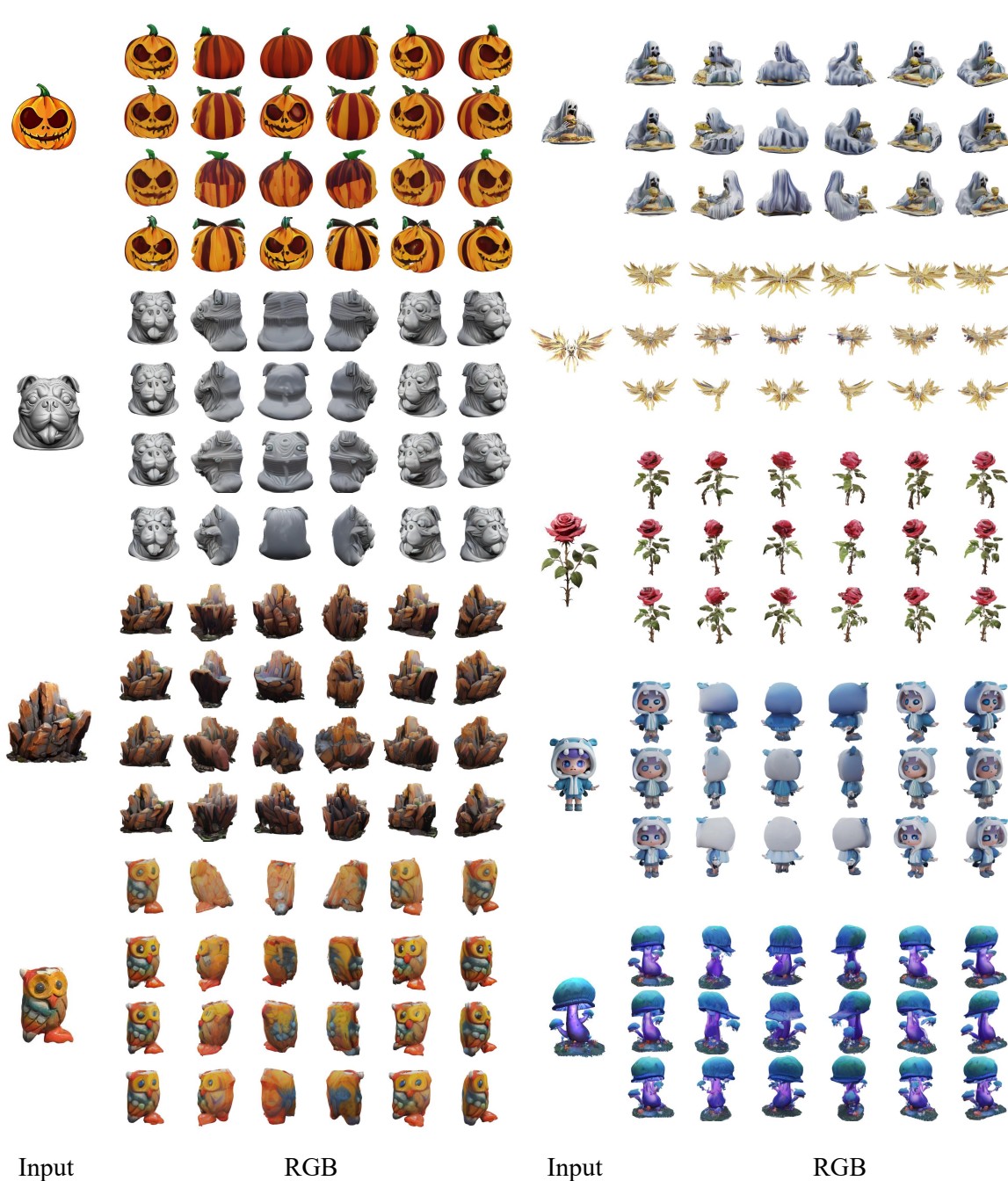

Figure 8: Generative 3D model with various geometry and texture given the same condition image, which shows strong generative ability of our model.

