# OpenReview forum: "ZERO-1-to-G: Taming Pretrained 2D Diffusion Models for Direct 3D Generation"
_ICLR.cc/2025/Conference — Submitted to ICLR 2025_

### Official Review · Reviewer_VjY6 · 2024-10-31

**Soundness:** 2
**Presentation:** 3
**Contribution:** 2
**Rating:** 5
**Confidence:** 5

**Summary:**

This paper introduces Zero-1-to-G, a novel 3D generative framework that generates 3D Gaussian splats from a single input image. The framework decomposes the generation process into five distinct attribute generations for fixed camera viewpoints, allowing a fine-tuned 2D diffusion model and a VAE decoder to generate multi-view, multi-attribute images.

**Strengths:**

* Decomposing the 3D Gaussian splat generation problem into multi-view and multi-attribute image generation is interesting.
* The paper is structured well, making it easy to understand and follow.

**Weaknesses:**

**Weakness 1**

The primary concern is the novelty of the paper. LGM [1] is designed to quickly generate Gaussian splats from various multi-view images, with the advantage of compatibility with different multi-view diffusion models. So, I think Zero-1-to-G can be regarded as a combination of an existing multi-view diffusion model and LGM, extending the cross-domain attention mechanism of Wonder3D [2] to five attributes.

**Weakness 2**

Experimental results are not sufficient to prove the effectiveness of Zero-1-to-G. Many One-Image-to-3D [2, 3, 4, 5] models report 3D reconstruction results (e.g. Chamfer Distance) on the GSO dataset. Since Zero-1-to-G takes 8.7 seconds to generate a 3D Gaussian splat, it would be better to show the performance with fewer sampling steps and a similar inference time to LGM.

[1] Tang et al., LGM: Large Multi-View Gaussian Model for High-Resolution 3D Content Creation, ECCV 2024.

[2] Long et al., Wonder3D: Single Image to 3D using Cross-Domain Diffusion, CVPR 2024.

[3] Woo et al., HarmonyView: Harmonizing Consistency and Diversity in One-Image-to-3D, CVPR 2024.

[4] Liu et al., SyncDreamer: Generating Multiview-consistent Images from a Single-view Image, ICLR 2024.

[5] Xu et al., DMV3D: DENOISING MULTI-VIEW DIFFUSION USING 3D LARGE RECONSTRUCTION MODEL, ICLR 2024.

**Questions:**

See weakness

---

> ### Author Response · Authors · 2024-11-23
> **In Reply to Official Review of Submission1229 by Reviewer VjY6**
>
> Thank Reviewer VjY6 for the careful review and valuable comments. We will address each weakness below.
>
> > The primary concern is the novelty of the paper. LGM [1] is designed to quickly generate Gaussian splats from various multi-view images, with the advantage of compatibility with different multi-view diffusion models. So, I think Zero-1-to-G can be regarded as a combination of an existing multi-view diffusion model and LGM, extending the cross-domain attention mechanism of Wonder3D [2] to five attributes.
>
> As noted by Reviewer cXYS and Reviewer wYEo, the ***novelty*** of our framework lies in effectively ***leveraging pretrained generative priors for 3D generation***. While some components and techniques in our method are inspired by prior works, our contribution is not a mere combination of existing ideas. Instead, we carefully integrate these components to maximize the utility of pretrained 2D priors for 3D generation
>
> Zero-1-to-G differentiates itself from prior multi-view methods such as LGM and Wonder3D in the following points:
> 1) ***addressing the limitations of their two-stage pipelines***: which involve multi-view generation followed by regressive reconstruction. This two-stage approach often suffers from multi-view inconsistencies in the first stage, which propagate errors into the 3D reconstruction. Zero-1-to-G overcomes this bottleneck with a single-stage design, directly learning a 3D generation process.
> 2) ***directly learning a distribution of 3D***: unlike previous methods that approximate a joint distribution of multi-view observations, Zero-1-to-G learns the true distribution of 3D representations, ensuring consistent 3D generation results. These innovations collectively represent a significant advancement in addressing challenges in multi-view and 3D generation.
>
> > Experimental results are not sufficient to prove the effectiveness of Zero-1-to-G. Many One-Image-to-3D [2, 3, 4, 5] models report 3D reconstruction results (e.g. Chamfer Distance) on the GSO dataset.
>
> We have evaluated Chamfer Distance on both our method and the baseline methods (on GSO dataset), and we will include these results in our revised submission:
>
> | Method     | Chamfer Distance     |
> |--------------|--------------|
> | LGM [1] | 0.0621 |
> | InstantMesh [2] | 0.0419 |
> | LN3Diff [3] | 0.0637 |
> | Triposr [4] | 0.0423 |
> |TriplaneGS [5] | 0.0440 |
> | Ours | **0.0390** |
>
> The improvement in Chamfer Distance further ***validates our claim that our approach generates superior 3D representations***, demonstrating the effectiveness of our method.
>
>
>
>
>
>
>
> *References*
>   [1] Tang et al., LGM: Large Multi-View Gaussian Model for High-Resolution 3D Content Creation, ECCV 2024.
>   [2] Xu et al. Instantmesh: Efficient 3d mesh generation from a single image with sparse-view large reconstruction models. arXiv preprint arXiv:2404.07191 (2024).
>   [3] Lan et al. Ln3diff: Scalable latent neural fields diffusion for speedy 3d generation." ECCV 2024.
>   [4] Tochilkin et al. Triposr: Fast 3d object reconstruction from a single image. arXiv 2024.
>   [5] Zou et al. Triplane meets gaussian splatting: Fast and generalizable single-view 3d reconstruction with transformers. CVPR 2024.

---

> > ### Comment · Reviewer_VjY6 · 2024-11-25
> > **Official comment by Reviewer VjY6**
> >
> > Thank you for clarifying my concerns about the novelty and the 3D reconstruction results. I carefully read the author's response, but still not sure about the novelty of this work. It appears that Zero-1-to-G relies on both a multi-view generation model and a large reconstruction model to construct its single-stage pipeline. So, I think this approach requires prior training for generating the five attributes (which means a two-stage pipeline), resulting in Zero-1-to-G being restricted to the specific generation scenario. Considering its effectiveness and limited novelty, I will maintain my current score.

---

> ### Author Response · Authors · 2024-11-25
> **Authors reply to Reviewer VjY6**
>
> We sincerely appreciate your attention to our rebuttal and the opportunity to further clarify your concerns. However, it seems there may be some misunderstandings regarding our approach. We would like to address your claim that "Zero-1-to-G relies on both a multi-view generation model and a large reconstruction model to construct its single-stage pipeline," as we believe this assertion misrepresents our method. Specifically:
>
> -  **No “reliance on a multi-view generation model”**:
>
> Our method does ***not depend on any*** multi-view generation model. If the reviewer is referring to the creation of ground-truth splatter data, it is important to clarify that we use ground-truth renderings directly as multi-view inputs to a large reconstruction model. These renderings are deterministic outputs derived from the ground-truth data, not generated by any separate model. Therefore, this step should not be construed as reliance on a multi-view generation model.
>
>
> + **Use of the large reconstruction model**:
> 1) The requirement for prior training to generate the five attributes is simply an approach to obtain ***effective 3D supervision***. It is worth noting that ***other direct 3D methods also require obtaining ground-truth 3D data*** as supervision, such as GVGen [1] and GaussianCube [2]. Therefore, the reconstruction of 3D ground truth is not an "extra reliance" unique to our method but rather a common step in achieving robust 3D learning
>
> 2) The large reconstruction model we employ is purely regressive and deterministic. While we fine-tune this model on our training data, it is not used as part of a multi-stage pipeline. Instead, its role is to generate high-quality splatter representations from ground-truth data, which are then used to supervise the training of our single-stage pipeline. This ensures that the supervision aligns well with the intended generation tasks while avoiding the need for an additional generative stage.
>
>
> + **“being restricted to the specific generation scenario”**:
> We want to emphasize that our method is ***not constrained to object-level generation***. In fact, it can be ***extended to scene-level generation with minimal modifications***. We are currently working on such extensions for scene generation, where simple replacement of the dataset has yielded promising results without requiring significant adaptation. This demonstrates the wide applicability and generalization capability of our approach.
>
>
> We hope this clarification helps address your concerns regarding novelty and the design of our pipeline. Thank you again for your thoughtful review and consideration.
>
> *Reference*
> [1] He et al. Gvgen: Text-to-3d generation with volumetric representation. ECCV, 2024.
> [2] Zhang et al. GaussianCube: Structuring Gaussian Splatting using Optimal Transport for 3D Generative Modeling. arXiv 2024.

---

> ### Author Response · Authors · 2024-11-25
> **Authors reply to Reviewer VjY6 - part 2**
>
> In addition, regarding previous questioning of **efficiency** in
> > Since Zero-1-to-G takes 8.7 seconds to generate a 3D Gaussian splat, it would be better to show the performance with fewer sampling steps and a similar inference time to LGM.
>
> As the reviewer suggested, we have ***reduced the denoising steps from 50 to 10***, achieving inference times comparable to LGM and significantly faster than InstantMesh. As shown in the ***revised PDF (Table 1)***, our method ***still outperforms baselines*** across metrics even with fewer steps, demonstrating its efficiency in generating high-quality 3D representations with faster inference.
>
> | Method     | Avg. Inference Time (s) |
> |--------------|--------------|
> | LGM-*30 steps*[1] | 2.98 |
> | InstantMesh [2] | 10.11 |
> | LN3Diff [3] | 7.51 |
> | Ours-*50 steps* | 7.69 |
> | Ours-*30 steps* | 3.46 |
> | Ours-*10 steps* | 1.60 |
>
>
> Hope this would address the reviewer's concern about the efficiency of Zero-1-to-G, since when the sampling steps are reduced to 10 for a faster inference speed, we still outperform baseline methods .
>
>
>
> *References*
> [1] Tang et al., LGM: Large Multi-View Gaussian Model for High-Resolution 3D Content Creation, ECCV 2024.
> [2] Xu et al. Instantmesh: Efficient 3d mesh generation from a single image with sparse-view large reconstruction models. arXiv preprint arXiv:2404.07191 (2024).
> [3] Lan et al. Ln3diff: Scalable latent neural fields diffusion for speedy 3d generation." ECCV 2024.

---

> ### Author Response · Authors · 2024-11-27
> **Official comment by Reviewer VjY6 -- part 3**
>
> We would like to further address Reviewer VjY6's concerns regarding the ***novelty*** of our approach:
>
> 1.  **Addressing Data and Training Efficiency:**
> Existing ***direct 3D generation*** methods (e.g., Shap-E [1], GVGen [2], GaussianCube [3] ) rely on 3D representations like point clouds or 3D Gaussian volumes, which require ***extensive data to train from scratch***, yet often fail to achieve high-quality details or generalize to in-the-wild data. Our method overcomes this limitation by leveraging the ***pretrained 2D diffusion model prior***.
>
> 2.  **Novel Decomposition for 2D Priors:**
> We propose decomposing splatter images into attribute images and have verified that they are aligned with the pretrained 2D model distribution. This allows our method to incorporate supervision from Internet-scale 2D data, significantly enhancing ***generalization*** to diverse scenarios. Metrics in *Table 1* (PSNR, LPIPS, SSIM) confirm the effectiveness of this approach.
>
> 3. **Improved 3D Consistency:**
> Compared to ***two-stage*** methods that use 2D priors for multi-view generation followed by reconstruction (e.g., LGM [4], InstantMesh [5] ), our single-stage pipeline ensures ***better 3D consistency and superior geometry quality***. This is evident from the improved Chamfer Distance results in *Table 1*.
>
> These points highlight the novelty of our method in achieving efficient, high-quality, and generalizable direct 3D generation.
> If the reviewer is still unsure, we are willing to address your concern further. Thank you for your feedback!
>
>
>
>
>
>
> *References*
> [1] Jun, Heewoo, and Alex Nichol. "Shap-e: Generating conditional 3d implicit functions." arXiv 2023.
> [2] He, Xianglong, et al. "Gvgen: Text-to-3d generation with volumetric representation." ECCV 2024.
> [3]  Zhang, Bowen, et al. "GaussianCube: Structuring Gaussian Splatting using Optimal Transport for 3D Generative Modeling." arXiv 2024.
> [4] Tang, Jiaxiang, et al. "Lgm: Large multi-view gaussian model for high-resolution 3d content creation." ECCV 2024.
> [5] Xu, Jiale, et al. "Instantmesh: Efficient 3d mesh generation from a single image with sparse-view large reconstruction models." arXiv 2024.

---

> ### Author Response · Authors · 2024-11-27
> **Additional Response to Reviewer**
>
> We sincerely appreciate your valuable feedback, which has been instrumental in improving our submission. Please let us know if there are any other concerns; we will try our best to address them. If not, please consider reassessing this work. We thank you for your patience and support. Thanks!

---

> > ### Comment · Reviewer_VjY6 · 2024-11-28
> > **Official comment by Reviewer VjY6**
> >
> > Thank you for the detailed explanation. I agree that my claim about the reliance on a multi-view diffusion model was incorrect, and I now understand the dependency lies solely on the reconstruction model, which is a common approach in previous methods that utilize additional 3D data.
> >
> > Once more, I believe Zero-1-to-G is an effective approach for generating 3D objects. However, I still think it primarily optimizes previous methods rather than presenting scientific motivations or novel concepts. For this reason, I find the novelty of Zero-1-to-G to be somewhat limited.
> >
> > It would be helpful if the authors could demonstrate that Zero-1-to-G is also effective for 3D scene generation, as they claim, and consistently provides advantages across various generation scenarios. Such evidence could outweigh the concerns about limited novelty. However, I will maintain my current score, as I cannot evaluate these aspects based on this submission.

---

> > > ### Author Response · Authors · 2024-12-02
> > >
> > > > Zero-1-to-G primarily optimizes previous methods rather than presenting scientific motivations or novel concepts
> > >
> > > We respectfully disagree with this assessment. Our work ***does not optimize previous methods*** but instead introduces a ***novel approach*** that is clearly different from previous 3D generation methods:
> > >
> > > + **Compared with two-stage methods:**
> > >   + our method ***directly*** generates 3D Gaussian and achieves better 3D consistency, while
> > >   + previous two-stage methods ***first generate multi-view*** images from single-view inputs and ***then use a reconstruction*** model to reconstruct 3D. The reconstructed 3D can have bad geometry due to inconsistent multi-view image generation.
> > >
> > > + **Compared with direct 3D generation:**
> > >    + our method leverages pretrained ***2D diffusion priors*** for direct 3D generation to achieve better generalizability, while
> > >    + previous direct 3D generation approaches need to train a network ***from scratch*** using only synthetic data, thus having ***limited generalizability*** to in-the-wild inputs.
> > >
> > > Therefore, regarding novelty, **Zero-1-to-G** is the ***first image-to-3D generative model that utilizes 2D diffusion priors to directly generate complete 3D representations***.
> > >
> > > > the novelty of Zero-1-to-G to be somewhat limited.
> > >
> > > We would like to further emphasize that adapting 2D pretrained diffusion models for direct 3D generation is a ***non-trivial*** problem. It requires selecting a *suitable 3D representation* and *effectively integrating it into existing 2D frameworks*. We made the following efforts to achieve this:
> > > 1)  **Alignment with pretrained latent spaces:**
> > >  We demonstrate that splatter-based 3D representations should be aligned with the latent space of pretrained Stable Diffusion to fully leverage its pretrained priors,
> > > 2) **Preserving rendering quality:**
> > >     To prevent degradation in the rendering quality of splatters after VAE decoding, we finetuned the VAE decoder using a rendering loss, ensuring high-fidelity outputs,
> > > 3) **Ensuring 3D consistency:**
> > >   To maintain consistency across 3D attributes and enable effective communication among all splatter attributes, we incorporated multi-view and multi-attribute attention mechanisms during the diffusion process.
> > >
> > > We hope the above clarification on our novelty and technical contribution could clearly address your concern, and we welcome your valuable feedback.

---

### Official Review · Reviewer_cXYS · 2024-11-02

**Soundness:** 3
**Presentation:** 3
**Contribution:** 2
**Rating:** 6
**Confidence:** 4

**Summary:**

This paper presents a novel method for 3d generation by finetuning a pretrained 2D diffusion model to generate 3D gaussian splatter images directly according to a single-view input image. The method integrates the generative priors from the pretrained diffusion model into the task of direct 3D generation and achieve good results.

**Strengths:**

* The paper proposes a novel framework for 3D generation by finetuning a pretrained 2D diffusion model for directly 3D generation, which is novel to me.
* The experiments are comprehensive and well-executed.
* The paper is well-written and easy to understand for people familiar with 3D generation tasks.

**Weaknesses:**

* The paper adopts LGM to build to overall dataset. This raises the concern of the overall quality of the training dataset. In line 216-217, the author declares that fitting-based reconstruction methods lead to excessively high-frequency signals which is hard for pretrained VAE to decode. However, in the paper, the author actually finetuned the decoder part of VAE with LGM outputs, which is confused. Why we could not finetune a VAE with the  fitting-based reconstruction data? The author should clarify this.
* The discuss of Limitation is not included in the paper.
* Overall, representing and generating Gaussian as images have been proposed by Splatter Image(CVPR2024), so the technological contribution of this paper is relatively limited. But I like the idea of finetuning a diffusion model to enjoy the generative priors, so I vote to a borderline accept.

[1] Splatter Image: Ultra-Fast Single-View 3D Reconstruction (https://arxiv.org/abs/2312.13150

**Questions:**

See Weaknesses.

---

> ### Author Response · Authors · 2024-11-23
> **In Reply to Official Review of Submission1229 by Reviewer cXYS - part 1**
>
> Thank Reviewer cXYS for acknowledging the strengths of our work and providing valuable feedback. We are more than willing to address each concern raised in your comments.
>
> > The paper adopts LGM to build to overall dataset. This raises the concern of the overall quality of the training dataset.
>
> Thank you for raising concerns about the potential limitations of using LGM-generated Splatter Image representations. In practice, we have not found this to be an issue, as detailed below:
>
> - **Bypassing Multiview Inconsistencies in LGM:**
>   Our approach builds on the reconstruction network (i.e., the second stage) of LGM, bypassing the first stage, which introduces multiview inconsistencies and represents the main performance bottleneck in LGM.
>
> - **Finetuning the Reconstruction Network:**
>   To further improve performance, we finetuned the reconstruction network on our training data. This introduced additional supervision from ground truth (GT) images, ensuring that our approach is not limited by the inference performance of LGM.
>
> - **Exploring Rendering Loss Supervision:**
>   To directly address concerns about potential limitations from LGM, we experimented with adding a rendering loss as additional supervision during diffusion training. This loss was supervised by ground-truth images following the second stage of training. However, we observed no significant performance improvements. This indicates that, at least given the current scale of training data and computational resources, the quality of Splatter Images is not a limiting factor for our method.
>
> > In line 216-217, the author declares that fitting-based reconstruction methods lead to excessively high-frequency signals which is hard for pretrained VAE to decode. However, in the paper, the author actually finetuned the decoder part of VAE with LGM outputs, which is confused. Why we could not finetune a VAE with the fitting-based reconstruction data? The author should clarify this.
>
> Thank you for raising this concern, and we apologize for any confusion. It is not contradictory to choose LGM’s output splatter while finetuning the decoder. The goal of finetuning is ***not to map to a new distribution*** but to enable the decoder to recover 3D information from a splatter image that is ***already within the pretrained VAE distribution*** (i.e., LGM’s output). This ensures we can ***leverage the pretrained prior*** for efficient training and improved generalization. A detailed comparison between fitting-based splatters and LGM’s output is provided below.
>
> - **Fitting-based splatter**
>    Finetuning on fitting-based splatter images often ***disrupts the pretrained latent space***, preventing effective use of the pretrained diffusion prior. Fitting methods introduce high-frequency artifacts, as shown in Figure 3 of our paper, due to the lack of constraints on Gaussian point arrangement. This excessive flexibility creates patterns that do not align with the smoother distributions pretrained VAEs are designed to handle, ultimately requiring significant changes to the decoder weights during fine-tuning.
> - **LGM’s output splatter**
>   In contrast, LGM’s output ***aligns better*** with the distribution of natural images. By using a UNet with skip connections, LGM imposes structure on the splatter images, making them more suitable for the pretrained VAE. With a stronger starting point for the VAE reconstruction, finetuning the decoder helps to further ***improve the rendering quality*** of the decoded splatters.
>
> Based on this insight, we argue that LGM's output splatters are a more effective choice than fitting-based splatters, as they more closely resemble natural images and are better aligned with the pretrained diffusion model's distribution.
>
> > The discuss of Limitation is not included in the paper.
>
> Thanks for pointing out this, we have added the below limitations in our revised supplementary *section A.2*.
>
> **Limitations** Despite achieving superior reconstruction metrics and strong generalization to in-the-wild data, our method has some limitations. First, our inference speed is not as fast as regression models, as each splatter must be generated through diffusion. A potential improvement would be to integrate a consistency model~\cite{song2023consistency} to reduce denoising steps and enhance inference speed. Second, we do not currently disentangle material and lighting conditions, leading to highlights and reflections being baked into the Gaussian splat texture. Future work could address this by incorporating inverse rendering to better predict non-Lambertian surfaces.

---

> ### Author Response · Authors · 2024-11-23
> **In Reply to Official Review of Submission1229 by Reviewer cXYS - part 2**
>
> > Overall, representing and generating Gaussian as images have been proposed by Splatter Image(CVPR2024), so the technological contribution of this paper is relatively limited. But I like the idea of finetuning a diffusion model to enjoy the generative priors, so I vote to a borderline accept.
>
> Thank you for recognizing the core insight of our work—leveraging 3D splatter images to utilize the prior knowledge of pretrained 2D diffusion models. While the concept of Gaussian splatter representations has been explored in prior work [1], our key contribution lies in the ***novel idea of leveraging pretrained 2D diffusion models for generating splatter images***, where it significantly enhances training efficiency and generalization. By utilizing generative priors, we address the challenges of limited 3D data and leverage the vast scale of internet data for improved 3D generation. We believe this perspective offers valuable guidance for future work in this field. Furthermore, our simple yet flexible network design enables easy adaptation and extension.
>
> Therefore, our application and improvement on [1] is non-trivial.
>
> [1] Splatter Image: Ultra-Fast Single-View 3D Reconstruction (https://arxiv.org/abs/2312.13150)

---

> ### Author Response · Authors · 2024-11-25
> **Additional Reply to Reviewer cXYS**
>
> We sincerely appreciate your valuable feedback, which has been instrumental in improving our submission.
>
> In response to your concerns, we have conducted additional experiments and expanded the discussions to address the questions you raised. We hope these updates adequately resolve your concerns and further enhance the quality of this work.
>
> As the deadline for the discussion period is approaching, we would greatly value any additional comments or suggestions to refine this work and meet your expectations. We kindly request you to reassess your score in light of this additional evidence. Thank you again for your time and thoughtful consideration.

---

> > ### Comment · Reviewer_cXYS · 2024-11-26
> >
> > Thanks for clarify my concern. I will maintain my score.

---

> > > ### Author Response · Authors · 2024-12-02
> > > **Regarding overall dataset quality**
> > >
> > > Thank you for raising concerns about the **overall quality of the training dataset**. To address this, we conducted a comparative analysis of ***our splatter ground truth*** *v.s.* that obtained from ***fitting-based*** methods, showcasing both *quality* and *efficiency*. Specifically, we evaluated the performance on the GSO dataset, which is unseen in our finetuned reconstruction model. For the fitting-based baseline, we rendered an equivalent number of training views for each object as used in our training dataset. The comparison results are as follows:
> > >
> > >
> > > | Method           | PSNR  | Time (Avg)        |
> > > |------------------|--------|-------------------|
> > > | Fitting-based    | 30.11  | 6 min 44 s        |
> > > | LGM Output       | 29.34  | 1.48 s *           |
> > >
> > > *( *Note: The time for LGM output can be further reduced with data parallelism, which does not apply to fitting-based methods due to the limitations of 3DGS optimization.
> > > "Time (Avg)" of both methods are tested on a single L40 GPU.
> > > )*
> > >
> > > Even accounting for the LGM model finetuning time (which can be as short as 1.5 hours on Objaverse with 8 L40 GPUs), our method remains significantly more efficient. Meanwhile, our method produces splatter ground truth that ***aligns well with the pretrained VAE distribution*** (please refer to Fig. 3 in the main paper). Importantly, this efficiency is achieved without compromising quality, as demonstrated by the comparable PSNR values.
> > >
> > > We sincerely appreciate your suggestion and tried our best to answer your questions. Please also consider reassessing the score if our explanation addresses your concerns sufficiently. Thank you for your time and thoughtful feedback.

---

### Official Review · Reviewer_wYEo · 2024-11-02

**Soundness:** 3
**Presentation:** 3
**Contribution:** 3
**Rating:** 6
**Confidence:** 5

**Summary:**

This paper introduces a new approach called ZERO-1-TO-G for 3D generation using Gaussian splats through 2D diffusion models. The key contribution of this paper is to represent 3D objects as multi-view, pixel-aligned Gaussian images and to fine-tune the VAE decoder so that the 3D objects can be generated directly. Experiments on GSO demonstrate promising results, with the proposed method outperforming several baselines in image-conditioned 3D generation tasks.

**Strengths:**

1. Due to the scarcity of 3D datasets, this paper presents a method to unleash the power of pretrained 2D diffusion models for direct 3D generation, with the potential to further enhance the performance of 3D generation.
2. The paper is generally well-written and easy to follow. The authors clearly explain the motivation behind their work, and the key components of their proposed method. The figures are helpful in illustrating the overall fine-tune VAE+LDM pipeline.
3. The experimental results show better performance than baselines.
4. The paper includes relevant ablations validating the individual design choices.

**Weaknesses:**

1. My main concern is that, despite the use of cross-view and cross-attribute attention layers to enhance 3D consistency, the generated results still exhibit inconsistencies across multiple views. How can we address these inconsistencies in the multi-view Gaussian splat results?
2. Methods such as 3DShape2Vec and 3DILG conduct experiments on shape autoencoding, which serves as the upper bound for the generated 3D objects. I believe that this experiment is also critical for ZERO-1-TO-G.
3. Missing comparisons experiment of generated 3d models for text-to-3d.
4. Missing reference:
[1] Cat3d: Create anything in 3d with multi-view diffusion models. NIPS 2024
[2] GPLD3D: Latent Diffusion of 3D Shape Generative Models by Enforcing Geometric and Physical Priors. CVPR 2024

**Questions:**

Please refer to the weaknesses above.

---

> ### Author Response · Authors · 2024-11-23
> **In Reply to Official Review of Submission1229 by Reviewer wYEo**
>
> Thank Reviewer wYEo for emphasizing the strengths of our work and providing valuable feedback. We are more than willing to address each concern raised in your comments.
>
> > My main concern is that, despite the use of cross-view and cross-attribute attention layers to enhance 3D consistency, the generated results still exhibit inconsistencies across multiple views. How can we address these inconsistencies in the multi-view Gaussian splat results?
>
> Our work ensures 3D consistency by ***learning a distribution of 3D representations*** (i.e., splatter images) through diffusion, rather than relying solely on cross-view and cross-attribute attention mechanisms. Below is a detailed explanation:
>
> - **True 3D Representation**
> Unlike methods that rely on partial 2D views, our multi-view splatter images represent the entire 3D scene. This enables the model to learn a true 3D distribution, as opposed to other approaches that only approximate a joint distribution of partial 2D projections. This comprehensive representation ensures better consistency across views.
>
> - **Relaxed Pixel Alignment**
> Multi-view RGB-based methods demand strict pixel alignment, which can introduce errors. In contrast, our multi-view splatter images feature a one-to-many mapping from 3D to 2D, allowing for greater flexibility. This relaxed alignment accommodates redundancies and ensures consistency without the constraints of rigid alignment.
>
> - **Loss Propagation Across Views**
> Our loss function propagates across all views simultaneously, ensuring global 3D consistency. Traditional multi-view methods, by contrast, apply losses to individual views, which can result in inconsistencies across views.
>
> - **Experimental Validation**
> Extensive experiments validate the superiority of our approach in achieving 3D consistency and 3D plausibility compared to other multi-view methods. (see qualitative results in Figure 4 of our paper).
>
> > Methods such as 3DShape2Vec and 3DILG conduct experiments on shape autoencoding, which serves as the upper bound for the generated 3D objects. I believe that this experiment is also critical for ZERO-1-TO-G.
>
> We have included the experiment to test the upper bound of our ground-truth 3D representation, which achieves ***PSNR=27.04*** and ***LPISP=0.042*** on GSO validation set used in our quantitative comparison. Qualitative and quantitative results are included in the supplementary of revision.
>
> > Missing comparisons experiment of generated 3d models for text-to-3d.
>
> Our task focuses on ***image-to-3D*** reconstruction, and our model is not trained with caption-3D pairs. Thank you for suggesting this direction; we will consider exploring it in future work.
>
>
> > Missing reference: [1] Cat3d: Create anything in 3d with multi-view diffusion models. NIPS 2024 [2] GPLD3D: Latent Diffusion of 3D Shape Generative Models by Enforcing Geometric and Physical Priors. CVPR 2024
>
> Thank you for pointing this out. We have added citations for these two papers in the revised version of our manuscript.

---

> > ### Comment · Reviewer_wYEo · 2024-11-28
> >
> > I have read the rebuttal carefully and would like to thank the authors. I greatly appreciate the authors for addressing my concern regarding the true 3D representation. I would like to maintain my original ranking.

---

> ### Author Response · Authors · 2024-11-25
> **Additional Response to Reviewer wYEo**
>
> We sincerely appreciate your valuable feedback, which has been instrumental in improving our submission.
>
> In response to your concerns, we have conducted additional experiments and expanded the discussions to address the questions you raised. We hope these updates adequately resolve your concerns and further enhance the quality of this work.
>
> As the deadline for the discussion period is approaching, we would greatly value any additional comments or suggestions to refine this work and meet your expectations. We kindly request you to reassess your score in light of this additional evidence. Thank you again for your time and thoughtful consideration.

---

### Official Review · Reviewer_k7Wt · 2024-11-03

**Soundness:** 3
**Presentation:** 4
**Contribution:** 2
**Rating:** 5
**Confidence:** 4

**Summary:**

This paper aims to leverage 2D priors in pre-trained image diffusion models for 3D content generation. The key idea is to adapt the previous work of Splatter Image as a proxy 2D representation of 3D information, such that multi-view image diffusion architectures can be applied.

**Strengths:**

1. The idea of leveraging 2D diffusion priors for 3D generation is promising, and the adaptation of Splatter Image structure is technically sound.
2. A series of correspondingly needed modifications are applied. For example, the different Gaussian attribute maps are normalized/reorganized to RGB space, and some self-attention operations are added for the modeling of both intra-view and inter-view distributions.

**Weaknesses:**

1. Generally, I tend to think that the proposed processing pipeline is a relatively simple and straightforward combination of existing techniques. First, the concept of Splatter Image is an existing work, and in this paper the adaptations are limited to some normalization and channel duplication operations. Second, the attention-related operations are also well-explored in previous methods, such as the "reshaping for attention" in MVDream and the "switcher mechanism" in Wonder3D.

2. I disagree with the authors' claim that "Zero-1-to-G is the first direct 3D generative model based on 2D diffusion frameworks". In terms of conceptual novelty and the target of leveraging 2D diffusion architectures, Omage [R1] and GIMDiffusion [R2] are even more representative works.
--[R1] X. Yan, et al., “An Object is Worth 64x64 Pixels: Generating 3D Object via Image Diffusion,” 2024.
--[R2] S. Elizarov, et al., “Geometry Image Diffusion: Fast and Data-Efficient Text-to-3D with Image-Based Surface Representation,” 2024.

3. I cannot see essential advantages of the proposed generation paradigm, because it still falls into the scope of multi-view image diffusion. Note that in the original design of Splatter Image, it can encode the whole 3D scene within a single image by modeling Gaussian position as a combination of depth and offset. Yet in this paper the Splatter Image still uses (x, y, z) coordinates and requires multi-view representations. Thus, the joint modeling of multi-view consistency is still a major challenge. Besides, the pre-trained diffusion priors are somewhat damaged, because the VAE component (the decoder) need to be retrained.

4. The ground-truth Splatter Image representations are generated by LGM, which fundamentally limiting the potential of the proposed method.

**Questions:**

Is it possible to use the single-image format of Splatter Image? In this way, the whole 3D scene can be encoded in one image-like tensor, thus avoiding the difficulty of constraining multi-view consistency. Of course, the VAE component also need to be retrained to adapt to the image-like tensor.

---

> ### Author Response · Authors · 2024-11-22
> **In reply to Official Review of Submission1229 by Reviewer k7Wt - part 1**
>
> We thank Reviewer k7Wt for all the valuable comments! We will address each point as below.
>
> > Generally, I tend to think that the proposed processing pipeline is a relatively simple and straightforward combination of existing techniques. First, the concept of Splatter Image is an existing work, and in this paper the adaptations are limited to some normalization and channel duplication operations. Second, the attention-related operations are also well-explored in previous methods, such as the "reshaping for attention" in MVDream and the "switcher mechanism" in Wonder3D.
>
> As Reviewer cXYS and Reviewer wYEo noted, the novelty of our framework lies in ***enabling 3D generation to leverage generative priors*** effectively. While some components and techniques in our method are inspired by prior works, our contribution is not a mere combination of existing ideas. Instead, we carefully integrate these components to maximize the utility of pretrained 2D priors for 3D generation.
>
> Our key insight is that splatter images—a 3D representation—can be decomposed into forms resembling real images, with ***each attribute image aligning well with the distribution of pretrained 2D diffusion models***. This enables us to leverage Stable Diffusion with minimal yet impactful modifications, resulting in a direct 3D diffusion model that produces high-quality 3D outputs. The simplicity of our pipeline reflects the strength of this principled approach rather than a lack of innovation.
>
> >  I disagree with the authors' claim that "Zero-1-to-G is the first direct 3D generative model based on 2D diffusion frameworks". In terms of conceptual novelty and the target of leveraging 2D diffusion architectures, Omage [R1] and GIMDiffusion [R2] are even more representative works. --[R1] X. Yan, et al., “An Object is Worth 64x64 Pixels: Generating 3D Object via Image Diffusion,” 2024. --[R2] S. Elizarov, et al., “Geometry Image Diffusion: Fast and Data-Efficient Text-to-3D with Image-Based Surface Representation,” 2024.
>
> We thank you for highlighting these two papers. We have included these references and revised the "first" claim in our updated manuscript. However, we kindly argue that our work remains the first to leverage a pretrained 2D diffusion model for direct single-view image-to-3D Gaussian generation. While the mentioned works are concurrent with ours (arXiv submissions), the key differences are discussed in the revised related work section and summarized below:
>
> 1. **Omage [R1]**:
>    This approach does not utilize ***pretrained*** diffusion priors and is trained on category-specific datasets, limiting its applicability to in-the-wild input samples.
>
> 2. **Geometry Image Diffusion [R2]**:
>    This approach only leverages pretrained diffusion priors for ***only texture*** generation. In contrast, our method use pretrained priors to geometry generation as well. Our key insight is that pretrained diffusion models can generate images beyond RGB textures, such as position maps, opacity maps, and scale maps. Using pretrained diffusion for the ***entire 3D*** generation process is central to our methodology and significantly distinguishes our work from others.

---

> ### Author Response · Authors · 2024-11-22
> **In reply to Official Review of Submission1229 by Reviewer k7Wt - part 2**
>
> > I cannot see essential advantages of the proposed generation paradigm, because it still falls into the scope of multi-view image diffusion. Note that in the original design of Splatter Image, it can encode the whole 3D scene within a single image by modeling Gaussian position as a combination of depth and offset. Yet in this paper the Splatter Image still uses (x, y, z) coordinates and requires multi-view representations. Thus, the joint modeling of multi-view consistency is still a major challenge.
>
> Thank you for this important question regarding the distinction between our work and previous multi-view image diffusion methods. We clarify the key differences in the following points:
>
> 1. **Comparison of Single-View vs. Multi-View Splatter Images**:
>  While a single splatter image does not suffer from potential multi-view inconsistencies, its ***representation capability*** is highly constrained compared to using multiple splatter images. The following table compares the PSNR values of single-view and multi-view splatter representations (on a validation set of Objaverse):
>
> | Splatter Type | PSNR(dB) |
> |----------|----------|
> | Single View (1 view) | 18.72 |
> | Multi-View (6 views) | 27.65 |
>
> *Note: The values here demonstrate a clear improvement in PSNR with the use of multi-view splatter images, confirming the enhanced representation power.*
>
> 2. **Global Coordinates (x, y, z) vs. Depth and Offset**:
>  Depth and offset are defined within the ***local*** coordinates of each view, while (x, y, z) are ***global*** coordinates, allowing for ***explicit correspondences*** across views and facilitating smoother information exchange between views. This results in more consistent multi-view representations.
>
> 3) **Ours (multi-view splatters) v.s. Other multi-view RGB methods**:
>  Ours does not fall into multi-view *image* inconsistency because we ***directly learn the distribution of 3D***, which will guarantee better 3D consistency although using similar multi-view generation process:
>
>     **a)** *Comprehensive 3D Representation*: Unlike other methods that rely on partial observations—training only on 2D projections from a limited subset of viewpoints—our multi-view splatter representation spans the entire 3D space. This is a critical distinction: our method learns a true 3D distribution, whereas other approaches approximate the joint distribution of partial 2D projections, which may lead to inconsistencies in unseen views.
>
>     **b)** *Loss Propagation and 3D Consistency*: Our loss function propagates across all views via the splatter image representation, ensuring 3D consistency throughout. This design inherently enforces global consistency across the views, maintaining a coherent 3D structure.
>
>     **c)** *Flexibility in Multi-View Consistency*: Traditional multi-view methods often rely on pixel-space alignment between multi-view RGB images, due to the unique and rigid mapping from 3D objects to 2D projections. In contrast, our method utilizes a one-to-many mapping from 3D objects to multi-view splatter images. This greater flexibility allows for more robust multi-view consistency during generation, as it is less prone to the rigid constraints of pixel alignment.
>
>
>
> > Besides, the pre-trained diffusion priors are somewhat damaged, because the VAE component (the decoder) need to be retrained.
>
> While the "forgetting"(i.e., damaging the prior) problem may arise under certain conditions, it does not apply to our work. As the splatter image is reconstructed with a high PSNR and with visually appealing through the finetuned decoder, which indicates that the mapping is ***well-preserved***. Below, we provide a detailed justification for our approach.
> 1) In fact, finetuning the VAE is a ***widely accepted*** practice for adapting to new data and objectives. For example, certain versions of Stable Diffusion finetune the VAE (both the encoder and decoder) using updated losses and datasets. Similarly, finetuning the VAE is a standard practice in video diffusion tasks to enhance frame-to-frame consistency. These adjustments are generally viewed not as "damaging" the prior, but rather as necessary adaptations to meet new requirements.
> 2) In our case, the pretrained VAE was ***initially*** trained to ***recover 2D images***, and it indeed can effectively reconstruct splatter images when viewed from a 2D perspective. The purpose of ***finetuning*** is to enable the VAE to better ***recover 3D information***, as it was not originally trained for splatter image reconstruction. This allows us to fully leverage the benefits of the pretrained prior, which serves as a strong starting point for training, as demonstrated in our ablation study.
>
> We hope this explanation clarifies our rationale and shows that finetuning is an intentional and effective strategy aligned with our objectives.

---

> ### Author Response · Authors · 2024-11-22
> **In reply to Official Review of Submission1229 by Reviewer k7Wt - part 3**
>
> > The ground-truth Splatter Image representations are generated by LGM, which fundamentally limiting the potential of the proposed method.
>
> Thank you for raising concerns about the potential limitations of using LGM-generated Splatter Image representations. In practice, we have not found this to be an issue, as detailed below:
>
> - **Bypassing Multiview Inconsistencies in LGM:**
>   Our approach builds on the reconstruction network (i.e., the second stage) of LGM, bypassing the first stage, which introduces multiview inconsistencies and represents the main performance bottleneck in LGM.
>
> - **Finetuning the Reconstruction Network:**
>   To further improve performance, we finetuned the reconstruction network on our training data. This introduced additional supervision from ground truth (GT) images, ensuring that our approach is not limited by the inference performance of LGM.
>
> - **Quality of Reconstructed Splatter Images:**
>   In practice, we find that the reconstructed splatter images from the second stage of LGM are of sufficiently high quality to serve as training data, with inference results on GSO dataset achieving *PSNR=27.04, LPIPS=0.042*, while ours achieves *PSNR=19.40, LPIPS=0.178*, thus not bounded.
>
> - **Exploring Rendering Loss Supervision:**
>   To directly address concerns about potential limitations from LGM, we experimented with adding a rendering loss as additional supervision during diffusion training. This loss was supervised by ground-truth images following the second stage of training. However, we observed no significant performance improvements. This indicates that, at least given the current scale of training data and computational resources, the quality of Splatter Images is not a limiting factor for our method.
>
> We hope this explanation clarifies that our approach effectively addresses potential concerns related to LGM and ensures robust performance.

---

> ### Author Response · Authors · 2024-11-23
> **In reply to Official Review of Submission1229 by Reviewer k7Wt - part 4**
>
> > Is it possible to use the single-image format of Splatter Image? In this way, the whole 3D scene can be encoded in one image-like tensor, thus avoiding the difficulty of constraining multi-view consistency. Of course, the VAE component also need to be retrained to adapt to the image-like tensor.
>
> Though the single-view format of Splatter Images avoids multi-view inconsistencies, its rendering quality is significantly lower due to limited capacity. Detailed analysis and results supporting this observation can be found in our response to Reviewer k7Wt (https://openreview.net/forum?id=nmc9ujrZ5R&noteId=Qvt2RVeK6R), including the PSNR comparisons of different splatter types.
>
> Based on this justification, multi-view splatter remains the better choice for achieving higher-quality results.

---

> ### Author Response · Authors · 2024-11-25
> **Additional Reply to Reviewer k7Wt**
>
> We sincerely appreciate your valuable feedback, which has been instrumental in improving our submission.
>
> In response to your concerns, we have conducted additional experiments and expanded the discussions to address the questions you raised. We hope these updates adequately resolve your concerns and further enhance the quality of this work.
>
> As the deadline for the discussion period is approaching, we would greatly value any additional comments or suggestions to refine this work and meet your expectations. We kindly request you to reassess your score in light of this additional evidence. Thank you again for your time and thoughtful consideration.

---

> ### Author Response · Authors · 2024-11-27
> **Additional Response to Reviewer**
>
> We sincerely appreciate your valuable feedback, which has been instrumental in improving our submission. Please let us know if there are any other concerns; we will try our best to address them. If not, please consider reassessing this work. We thank you for your patience and support. Thanks!

---

> ### Author Response · Authors · 2024-12-01
> **Kind Reminder for Response**
>
> Dear Reviewer k7Wt,
>
> Thanks so much for your time and effort in reviewing our submission! We deeply appearciate it.
>
> We highly value any opportunities to address any potential remaining concerns before the discussion closes, which might be helpful for improving the rating of this submission. Please do not hesitate to comment upon any further concerns. Your feedback is extremely valuable!
>
> Thanks,
>
> Authors

---

### Author Response · Authors · 2024-11-23
**General Response**

We thank all the reviewers for their valuable feedback on our work. In response to common concerns, we emphasize that our main contribution lies in the ***novel insight*** that splatter images—a 3D representation—can be decomposed into forms resembling real images, with each attribute image aligning closely with the distribution of pretrained 2D diffusion models. This allows us to leverage Stable Diffusion with minimal yet impactful modifications, enabling a direct 3D diffusion model that achieves high-quality 3D generation.

Our contribution can be summarized in three key aspects:

1. **Leveraging Pretrained Priors**:
   We find that each attribute image of splatter image is modeled within the distribution of Stable Diffusion, therefore we propose to use the pretrained prior of Stable Diffusion as a good starting point to train our 3D generation model.

2. **Addressing Limited 3D Data**:
   Our method overcomes the challenge of limited 3D training data by utilizing pretrained 2D priors learned from large-scale internet data. This enables efficient training with strong generalization capabilities.

3. **Ensuring 3D Consistency**:
   Unlike multi-view 2D methods that rely on the joint distribution of multi-view images, our approach directly learns in a 3D latent space. This ensures superior 3D consistency compared to existing multi-view pipelines.

We hope this perspective inspires future research in this area. Furthermore, our modular and compatible network design provides flexibility for researchers looking to extend or build upon our approach.

---

### Meta-Review · Area_Chair_Sv9m · 2024-12-17

**Metareview:**

This paper introduces Zero-1-to-G, a framework leveraging 2D diffusion priors for 3D Gaussian splat generation. The proposed method shows promise in adapting 2D pre-trained diffusion models for 3D tasks; however, there are still concerns about its novelty and validation. The mentioned key contributions, such as fine-tuning a VAE decoder and multi-view attention mechanisms, are likely incremental extensions of prior works. Moreover, reliance on LGM-generated training data, and limited experimental comparisons with strong baselines further affect its impact. I’d recommend the authors to consider all these and further improve the work.

**Additional Comments On Reviewer Discussion:**

There are several rounds of discussion among authors and the reviewers. The major concern is about the novelty of this work. Multi-view consistency concern has been resolved via discussion. However, concerns about the novelty of the 2D diffusion models for 3D applications still remain.

---

### Decision · Program_Chairs · 2025-01-22

Reject